# Learning Stochastic Inverses

**Andreas Stuhlmüller**
Brain and Cognitive Sciences
MIT

**Jessica Taylor**
Department of Computer Science
Stanford University

**Noah D. Goodman**
Department of Psychology
Stanford University

## Abstract

We describe a class of algorithms for *amortized inference* in Bayesian networks. In this setting, we invest computation upfront to support rapid online inference for a wide range of queries. Our approach is based on learning an inverse factorization of a model's joint distribution: a factorization that turns observations into root nodes. Our algorithms accumulate information to estimate the local conditional distributions that constitute such a factorization. These *stochastic inverses* can be used to invert each of the computation steps leading to an observation, sampling *backwards* in order to quickly find a likely explanation. We show that estimated inverses converge asymptotically in number of (prior or posterior) training samples. To make use of inverses before convergence, we describe the *Inverse MCMC* algorithm, which uses stochastic inverses to make block proposals for a Metropolis-Hastings sampler. We explore the efficiency of this sampler for a variety of parameter regimes and Bayes nets.

## 1 Introduction

Bayesian inference is computationally expensive. Even approximate, sampling-based algorithms tend to take many iterations before they produce reasonable answers. In contrast, human recognition of words, objects, and scenes is extremely rapid, often taking only a few hundred milliseconds—only enough time for a single pass from perceptual evidence to deeper interpretation. Yet human perception and cognition are often well-described by probabilistic inference in complex models. How can we reconcile the speed of recognition with the expense of coherent probabilistic inference? How can we build systems, for applications like robotics and medical diagnosis, that exhibit similarly rapid performance at challenging inference tasks?

One response to such questions is that these problems are not, and should not be, solved from scratch each time they are encountered. Humans and robots are in the setting of *amortized inference*: they have to solve many similar inference problems, and can thus offload part of the computational work to shared precomputation and adaptation over time. This raises the question of which kinds of precomputation and adaptation are useful. There is substantial previous work on adaptive inference algorithms, including Cheng and Druzdzel (2000); Haario et al. (2006); Ortiz and Kaelbling (2000); Roberts and Rosenthal (2009). While much of this work is focused on adaptation for a single posterior inference, amortized inference calls for adaptation across many different inferences. In this setting, we will often have considerable training data available in the form of posterior samples from previous inferences; how should we use this data to adapt our inference procedure?

We consider using training samples to learn the *inverse* structure of a directed model. Posterior inference is the task of inverting a probabilistic model: Bayes' theorem turns $p(d|h)$ into $p(h|d)$; vision is commonly understood as inverse graphics (Horn, 1977) and, more recently, as inverse physics (Sanborn et al., 2013; Watanabe and Shimojo, 2001); and conditional inference in probabilistic programs can be described as "running a program backwards" (e.g., Wingate and Weber, 2013). However, while this is a good description of the problem that inference solves, conditional sampling usually does *not* proceed backwards step-by-step. We suggest taking this view more liter-

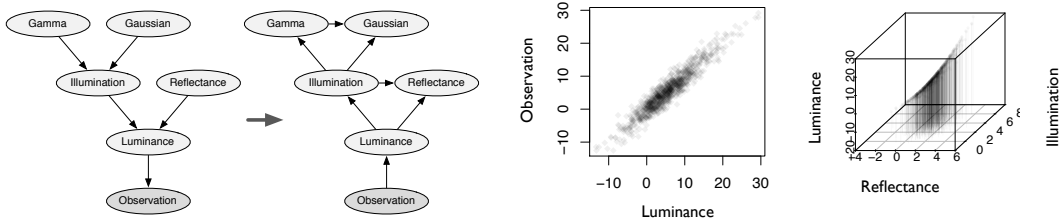

Figure 1: A Bayesian network modeling brightness constancy in visual perception, a possible inverse factorization, and two of the local joint distributions that determine the inverse conditionals.

ally and actually learning the inverse conditionals needed to invert the model. For example, consider the Bayesian network shown in Figure 1. In addition to the default "forward" factorization shown on the left, we can consider an "inverse" factorization shown on the right. Knowing the conditionals for this inverse factorization would allow us to rapidly sample the latent variables given an observation. In this paper, we will explore what these factorizations look like for Bayesian networks, how to learn them, and how to use them to construct block proposals for MCMC.

## 2   Inverse factorizations

Let $p$ be a distribution on latent variables $x = (x_1, \ldots, x_m)$ and observed variables $y = (y_1, \ldots, y_n)$. A Bayesian network $G$ is a directed acyclic graph that expresses a factorization of this joint distribution in terms of the distribution of each node conditioned on its parents in the graph:

$$p(x, y) = \prod_{i=1}^{m} p(x_i | \mathrm{pa}_G(x_i)) \prod_{j=1}^{n} p(y_j | \mathrm{pa}_G(y_j))$$

When interpreted as a generative (causal) model, the observations $y$ typically depend on a non-empty set of parents, but are not themselves parents of any nodes.

In general, a distribution can be represented using many different factorizations. We say that a Bayesian network $H$ expresses an *inverse factorization* of $p$ if the observations $y$ do not have parents (but may themselves be parents of some $x_i$):

$$p(x, y) = p(y) \prod_{i=1}^{m} p(x_i | \mathrm{pa}_H(x_i))$$

As an example, consider the forward and inverse networks shown in Figure 1. We call the conditional distributions $p(x_i | \mathrm{pa}_H(x_i))$ *stochastic inverses*, with *inputs* $\mathrm{pa}_H(x_i)$ and *output* $x_i$. If we could sample from these distributions, we could produce samples from $p(x|y)$ for arbitrary $y$, which solves the problem of inference for all queries with the same set of observation nodes.

In general, there are many possible inverse factorizations. For each latent node, we can find a factorization such that this node does not have children. This fact will be important in Section 4 when we resample subsets of inverse graphs. Algorithm 1 gives a heuristic method for computing an inverse factorization given Bayes net $G$, observation nodes $y$, and desired leaf node $x_i$. We compute an ordering on the nodes of the original Bayes net from observations to leaf node. We then add the nodes in order to the inverse graph, with dependencies determined by the graph structure of the original network.

In the setting of amortized inference, past tasks provide approximate posterior samples for the corresponding observations. We therefore investigate learning inverses from such samples, and ways of using approximate stochastic inverses for improving the efficiency of solving future inference tasks.

---
**Algorithm 1:** Heuristic inverse factorization
---
**Input:** Bayesian network $G$ with latent nodes $x$ and observed nodes $y$; desired leaf node $x_i$
**Output:** Ordered inverse graph $H$
 1: order $x$ such that nodes close to $y$ are first, leaf node $x_i$ is last
 2: initialize $H$ to empty graph
 3: add nodes $y$ to $H$
 4: **for** node $x_j$ **in** $x$ **do**
 5:     add $x_j$ to $H$
 6:     set $\mathrm{pa}_H(x_j)$ to a minimal set of nodes in $H$ that d-separates $x_j$ from the remainder of $H$ based on the graph structure of $G$
 7: **end for**
---

## 3  Learning stochastic inverses

It is easy to see that we can estimate conditional distributions $p(x_i|\mathrm{pa}_H(x_i))$ using samples $S$ drawn from the prior $p(x, y)$. For simplicity, consider discrete variables and an empirical frequency estimator:

$$\theta_S(x_i|\mathrm{pa}_H(x_i)) = \frac{|\{s \in S : x_i^{(s)} \wedge \mathrm{pa}_H^{(s)}(x_i)\}|}{|\{s \in S : \mathrm{pa}_H^{(s)}(x_i)|}$$

Because $\theta_S$ is a consistent estimator of the probability of each outcome for each setting of the parent variables, the following theorem follows immediately from the strong law of large numbers:

**Theorem 1.** (Learning from prior samples) *Let $H$ be an inverse factorization. For samples $S$ drawn from $p(x, y)$, $\theta_S(x_i|\mathrm{pa}_H(x_i)) \to p(x_i|\mathrm{pa}_H(x_i))$ almost surely as $|S| \to \infty$.*

Samples generated from the prior may be sparse in regions that have high probability under the posterior, resulting in slow convergence of the inverses. We now show that valid inverse factorizations allow us to learn from posterior samples as well.

**Theorem 2.** (Learning from posterior samples) *Let $H$ be an inverse factorization. For samples $S$ drawn from $p(x|y)$, $\theta(x_i|\mathrm{pa}_H(x_i)) \to p(x_i|\mathrm{pa}_H(x_i))$ almost surely as $|S| \to \infty$ for values of $\mathrm{pa}_H(x_i)$ that have positive probability under $p(x|y)$.*

*Proof.* For values $\mathrm{pa}_H(x_i)$ that are not in the support of $p(x|y)$, $\theta(x_i|\mathrm{pa}_H(x_i))$ is undefined. For values $\mathrm{pa}_H(x_i)$ in the support, $\theta(x_i|\mathrm{pa}_H(x_i)) \to p(x_i|\mathrm{pa}_H(x_i), y)$ almost surely. By definition, any node in a Bayesian network is independent of its non-descendants given its parent variables. The nodes $y$ are root nodes in $H$ and hence do not descend from $x_i$. Therefore, $p(x_i|\mathrm{pa}_H(x_i), y) = p(x_i|\mathrm{pa}_H(x_i))$ and the theorem holds. □

Theorem 2 implies that we can use posterior samples from one observation set to learn inverses that apply to all other observation sets—while samples from $p(x|y)$ only provide global estimates for the given posterior, it is guaranteed that the local estimates created by the procedure above are equivalent to the query-independent conditionals $p(x_i|\mathrm{pa}_H(x_i))$. In addition, we can combine samples from distributions conditioned on several different observation sets to produce more accurate estimates of the inverse conditionals.

In the discussion above, we can replace $\theta$ with any consistent estimator of $p(x_i|\mathrm{pa}_H(x_i))$. We can also trade consistency for faster learning and generalization. This framework can make use of any supervised machine learning technique that supports sampling from a distribution on predicted outputs. For example, for discrete variables we can employ logistic regression, which provides fast generalization and efficient sampling, but cannot, in general, represent the posterior exactly. Our choice of predictor can be data-dependent—for example, we can add interaction terms to a logistic regression predictor as more data becomes available.

For continuous variables, consider a predictor based on k-nearest neighbors that produces samples as follows (Algorithm 2): Given new input values $z$, retrieve the $k$ previously observed input-output

---

**Algorithm 2:** K-nearest neighbor density predictor

---

**Input:** Variable index $i$, inverse inputs $z$, samples $S$, number of neighbors $k$
**Output:** Sampled value for node $x_i$
  1: retrieve $k$ nearest pairs $(z^{(1)}, x_i^{(1)}), \ldots, (z^{(k)}, x_i^{(k)})$ in $S$ based on distance to $z$
  2: construct density estimate $q$ on $x_i^{(1)}, \ldots, x_i^{(k)}$
  3: sample from $q$

---

pairs that are closest to the current input values. Then, use a consistent density estimator to construct a density estimate on the nearby previous outputs and sample an output $x_i$ from the estimated distribution.

Showing that this estimator converges to the true conditional density $p(x|z)$ is more subtle. If the conditional densities are smooth in the sense that:

$$\forall \varepsilon > 0 \,\exists \delta > 0 : \forall z_1, z_2 \; d(z_1, z_2) < \delta \;\Rightarrow\; D_{\mathrm{KL}}(p(x|z_1), p(x|z_2)) < \varepsilon$$

then we can achieve any desired accuracy of approximation by assuring that the nearest neighbors used all lie within a $\delta$-ball, but that the number of neighbors goes to infinity. We can achieve this by increasing $k$ slowly enough in $|S|$. The exact rate at which we may increase depends on the distribution and may be difficult to determine.

## 4  Inverse MCMC

We have described how to compute the structure of inverse Bayes nets, and how to learn the associated conditional distributions and densities from prior and posterior samples. This produces fast, but possibly biased recognition models. To get a consistent estimator, we use these recognition models as part of a Metropolis-Hastings scheme that, as the amount of training data grows, converges to Gibbs sampling for proposals of size 1, to blocked-Gibbs for larger proposals, and to perfect posterior sampling for proposals of size $|G|$.

We propose the following *Inverse MCMC* procedure (Algorithm 3): *Offline*, use Algorithm 1 to compute an inverse graph for each latent node and train each local inverse in this graph from (posterior or prior) samples. *Online*, run Metropolis-Hastings with the proposal mechanism shown in Algorithm 4, which resamples a set of up to $k$ variables using the trained inverses[1]. With little training data, we will want to make small proposals (small $k$) in order to achieve a reasonable acceptance rate; with more training data, we can make larger proposals and expect to succeed.

**Theorem 3.** *Let $G$ be a Bayesian network, let $\theta$ be a consistent estimator (for inverse conditionals), let $\{H_i\}_{i \in 1..m}$ be a collection of inverse graphs produced using Algorithm 1, and assume a source of training samples (prior or posterior) with full support. Then, as training set size $|S| \to \infty$, Inverse MCMC with proposal size $k$ converges to block-Gibbs sampling where blocks are the last $k$ nodes in each $H_i$. In particular, it converges to Gibbs sampling for proposal size $k = 1$ and to exact posterior sampling for $k = |G|$.*

*Proof.* We must show that proposals are made from the conditional posterior in the limit of large training data. Fix an inverse $H$, and let $\mathbf{x}$ be the last $k$ variables in $H$. Let $\mathrm{pa}_H(\mathbf{x})$ be the union of $H$-parents of variables in $\mathbf{x}$ that are not themselves in $\mathbf{x}$. By construction according to Algorithm 1, $\mathrm{pa}_H(\mathbf{x})$ form a Markov blanket of $\mathbf{x}$ (that is, $\mathbf{x}$ is conditionally independent of other variables in $G$, given $\mathrm{pa}_H(\mathbf{x})$). Now the conditional distribution over $\mathbf{x}$ factorizes along the inverse graph: $p(\mathbf{x}|\mathrm{pa}_H(\mathbf{x})) = \prod_{i=k}^{|H|} p(x_i|\mathrm{pa}_H(x_i))$. But by theorems 1 and 2, the estimators $\theta$ converge, when they are defined, to the corresponding conditional distributions, $\theta(x_i|\mathrm{pa}_H(x_i)) \to p(x_i|\mathrm{pa}_H(x_i))$; since we assume full support, $\theta(x_i|\mathrm{pa}_H(x_i))$ is defined wherever $p(x_i|\mathrm{pa}_H(x_i))$ is defined. Hence, using the estimated inverses to sequentially sample the $\mathbf{x}$ variables results, in the limit, in samples from the conditional distribution given remaining variables. (Note that, in the limit, these proposals will always be accepted.) This is the definition of block-Gibbs sampling. The special cases of $k = 1$ (Gibbs) and $k = |G|$ (posterior sampling) follow immediately. ∎

**Algorithm 3:** Inverse MCMC

**Input:** Prior or posterior samples $S$
**Output:** Samples $x^{(1)}, \ldots, x^{(T)}$
*Offline (train inverses)*:
 1: **for** $i$ **in** $1 \ldots m$ **do**
 2:     $H_i \leftarrow$ from Algorithm 1
 3:     **for** $j$ **in** $1 \ldots m$ **do**
 4:         train inverse $\theta_S(x_j | \mathrm{pa}_{H_i}(x_j))$
 5:     **end for**
 6: **end for**
*Online (MH with inverse proposals)*:
 1: **for** $t$ **in** $1 \ldots T$ **do**
 2:     $x', p_{\mathrm{fw}}, p_{\mathrm{bw}}$ from Algorithm 4
 3:     $x \leftarrow x'$ with MH acceptance rule
 4: **end for**

**Algorithm 4:** Inverse MCMC proposer

**Input:** State $x$, observations $y$, ordered inverse graphs $\{H_i\}_{i \in 1..m}$, proposal size $k_{\max}$, inverses $\theta$
**Output:** Proposed state $x'$, forward and backward probabilities $p_{\mathrm{fw}}$ and $p_{\mathrm{bw}}$
 1: $H \sim \mathrm{Uniform}(\{H_i\}_{i \in 1..m})$
 2: $k \sim \mathrm{Uniform}(\{0, 1, \ldots, k_{\max} - 1\})$
 3: $x' \leftarrow x$
 4: $p_{\mathrm{fw}}, p_{\mathrm{bw}} \leftarrow 0$
 5: **for** $j$ **in** $n - k, \ldots, n$ **do**
 6:     let $x_l$ be $j$th variable in $H$
 7:     $x'_l \sim \theta(x_l | \mathrm{pa}_H(x'_l))$
 8:     $p_{\mathrm{fw}} \leftarrow p_{\mathrm{fw}} \cdot p_\theta(x'_l | \mathrm{pa}_H(x'_l))$
 9:     $p_{\mathrm{bw}} \leftarrow p_{\mathrm{bw}} \cdot p_\theta(x_l | \mathrm{pa}_H(x_l))$
10: **end for**

Instead of learning the $k{=}1$ "Gibbs" conditionals for each inverse graph, we can often precompute these distributions to "seed" our sampler. This suggests a bootstrapping procedure for amortized inference on observations $y^{(1)}, \ldots, y^{(t)}$: first, precompute the "Gibbs" distributions so that $k{=}1$ proposals will be reasonably effective; then iterate between training on previously generated approximate posterior samples and doing inference on the next observation. Over time, increase the size of proposals, possibly depending on acceptance ratio or other heuristics.

For networks with near-deterministic dependencies, Gibbs may be unable to generate training samples of sufficient quality. This poses a chicken-and-egg problem: we need a sufficiently good posterior sampler to generate the data required to train our sampler. To address this problem, we propose a simple annealing scheme: We introduce a temperature parameter $t$ that controls the extent to which (almost-)deterministic dependencies in a network are relaxed. We produce a sequence of trained samplers, one for each temperature, by generating samples for a network with temperature $t_{i+1}$ using a sampler trained on approximate samples for the network with next-higher temperature $t_i$. Finally, we discard all samplers except for the sampler trained on the network with $t = 0$, the network of interest.

In the next section, we explore the practicality of such bootstrapping schemes as well as the general approach of Inverse MCMC.

## 5 Experiments

We are interested in networks such that (1) there are many layers of nodes, with some nodes far removed from the evidence, (2) there are many observation nodes, allowing for a variety of queries, and (3) there are strong dependencies, making local Gibbs moves challenging.

We start by studying the behavior of the Inverse MCMC algorithm with empirical frequency estimator on a 225-node rectangular grid network from the UAI 2008 inference competition. This network has binary nodes and approximately 50% deterministic dependencies, which we relax to dependencies with strength .99. We select the 15 nodes on the diagonal as observations and remove any nodes below, leaving a triangular network with 120 nodes and treewidth 15 (Figure 2). We compute the true marginals $P^*$ using IJGP (Mateescu et al., 2010), and calculate the error of our estimates $P^s$ as

$$\text{error} = \frac{1}{N} \sum_{i=1}^{N} \frac{1}{|X_i|} \sum_{x_i \in X_i} |P^*(X_i = x_i) - P^s(X_i = x_i)|.$$

We generate 20 inference tasks as sources of training samples by sampling values for the 15 observation nodes uniformly at random. We precompute the "final" inverse conditionals as outlined above, producing a Gibbs sampler when $k{=}1$. For each inference task, we use this sampler to generate $10^5$ approximate posterior samples.

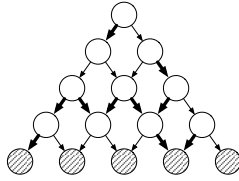

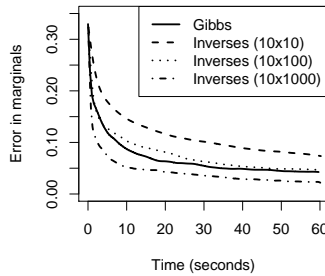

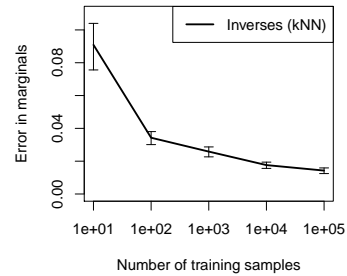

Figure 2: Schema of the Bayes net structure used in experiment 1. Thick arrows indicate almost-deterministic dependencies, shaded nodes are observed. The actual network has 15 layers with a total of 120 nodes.

Figure 3: The effect of training on approximate posterior samples for 10 inference tasks. As the number of training samples per task increases, Inverse MCMC with proposals of size 20 performs new inference tasks more quickly.

Figure 4: Learning an inverse distribution for the brightness constancy model (Figure 1) from prior samples using the KNN density predictor. More training samples result in better estimates after the same number of MCMC steps.

Figures 3 and 5 show the effect of training the frequency estimator on 10 inference tasks and testing on a different task (averaged over 20 runs). Inverse proposals of (up to) size $k=20$ do worse than pure Gibbs sampling with little training (due to higher rejection rate), but they speed convergence as the number of training samples increases. More generally, large proposals are likely to be rejected without training, but improve convergence after training.

Figure 6 illustrates how the number of inference tasks influences error and MH acceptance ratio in a setting where the total number of training samples is kept constant. Surprisingly, increasing the number of training tasks from 5 to 15 has little effect on error and acceptance ratio for this network. That is, it seems relatively unimportant which posterior the training samples are drawn from; we may expect different results when posteriors are more sparse.

Figure 7 shows how different sources of training data affect the quality of the trained sampler (averaged over 20 runs). As the strength of near-deterministic dependencies increases, direct training on Gibbs samples becomes infeasible. In this regime, we can still train on prior samples and on Gibbs samples for networks with relaxed dependencies. Alternatively, we can employ the annealing scheme outlined in the previous section. In this example, we take the temperature ladder to be $[.2, .1, .05, .02, .01, 0]$—that is, we start by learning inverses for the relaxed network where all CPT probabilities are constrained to lie within $[.2, .8]$; we then use these inverses as proposers for MCMC inference on a network constrained to CPT probabilities in $[.1, .9]$, learn the corresponding inverses, and continue, until we reach the network of interest (at temperature 0).

While the empirical frequency estimator used in the above experiments provides an attractive asymptotic convergence guarantee (Theorem 3), it is likely to generalize slowly from small amounts of training data. For practical purposes, we may be more interested in getting useful generalizations quickly than converging to a perfect proposal distribution. Fortunately, the Inverse MCMC algorithm can be used with any estimator for local conditionals, consistent or not. We evaluate this idea on a 12-node subset of the network used in the previous experiments. We learn complete inverses, resampling up to 12 nodes at once. We compare inference using a logistic regression estimator with $L_2$ regularization (with and without interaction terms) to inference using the empirical frequency estimator. Figure 9 shows the error (integrated over time to better reflect convergence speed) against the number of training examples, averaged over 300 runs. The regression estimator with interaction terms results in significantly better results when training on few posterior samples, but is ultimately overtaken by the consistent empirical estimator.

Next, we use the KNN density predictor to learn inverse distributions for the continuous Bayesian network shown in Figure 1. To evaluate the quality of the learned distributions, we take 1000

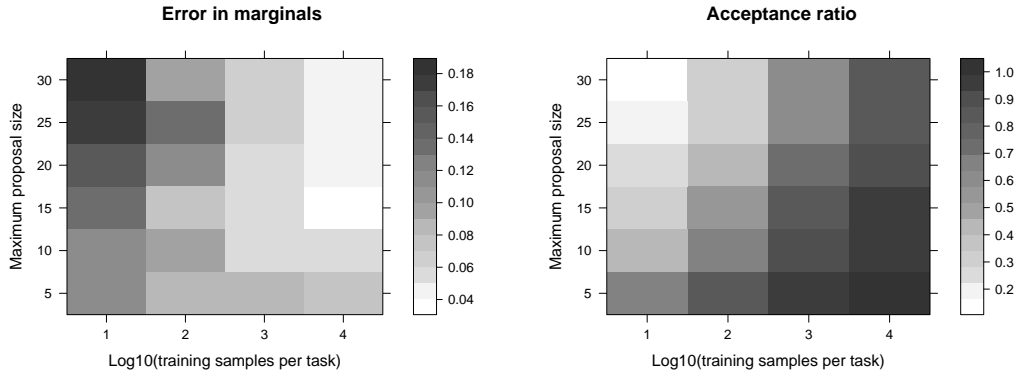

Figure 5: Without training, big inverse proposals result in high error, as they are unlikely to be accepted. As we increase the number of approximate posterior samples used to train the MCMC sampler, the acceptance probability for big proposals goes up, which decreases overall error.

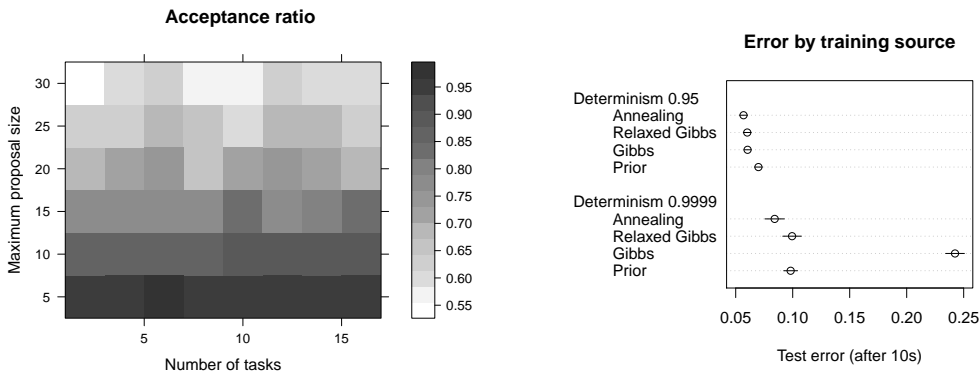

Figure 6: For the network under consideration, increasing the number of tasks (i.e., samples for other observations) we train on has little effect on acceptance ratio (and error) if we keep the total number of training samples constant.

Figure 7: For networks without hard determinism, we can train on Gibbs samples. For others, we can use prior samples, Gibbs samples for relaxed networks, and samples from a sequence of annealed Inverse samplers.

samples using Inverse MCMC and compare marginals to a solution computed by JAGS (Plummer et al., 2003). As we refine the inverses using forward samples, the error in the estimated marginals decreases towards 0, providing evidence for convergence towards a posterior sampler (Figure 4).

To evaluate Inverse MCMC in more breadth, we run the algorithm on all binary Bayes nets with up to 500 nodes that have been submitted to the UAI 08 inference competition (216 networks). Since many of these networks exhibit strong determinism, we train on prior samples and apply the annealing scheme outlined above to generate approximate posterior samples. For training and testing, we use the evidence provided with each network. We compute the error in marginals as described above for both Gibbs (proposal size 1) and Inverse MCMC (maximum proposal size 20). To summarize convergence over the 1200s of test time, we compute the area under the error curves (Figure 8). Each point represents a single run on a single model. We label different classes of networks. For the grid networks, grid-$k$ denotes a network with $k$% deterministic dependencies. While performance varies across network classes—with extremely deterministic networks making the acquisition of training data challenging—the comparison with Gibbs suggests that learned block proposals frequently help.

Overall, these results indicate that Inverse MCMC is of practical benefit for learning block proposals in reasonably large Bayes nets and using a realistic amount of training data (an amount that might result from amortizing over five or ten inferences).

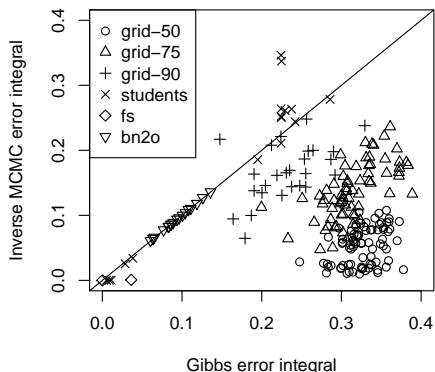

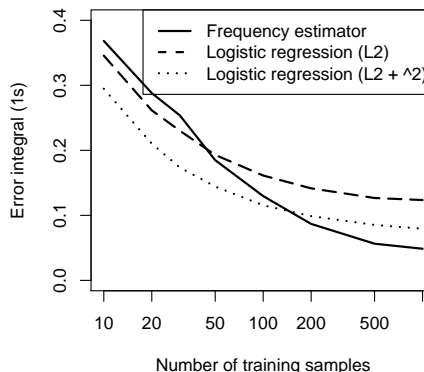

Figure 8: Each mark represents a single run of a model from the UAI 08 inference competition. Marks below the line indicate that integrated error over 1200s of inference is lower for Inverse MCMC than Gibbs sampling.

Figure 9: Integrated error (over 1s of inference) as a function of the number of samples used to train inverses, comparing logistic regression with and without interaction terms to an empirical frequency estimator.

## 6 Related work

A recognition network (Morris, 2001) is a multilayer perceptron used to predict posterior marginals. In contrast to our work, a single global predictor is used instead of small, compositional prediction functions. By learning local inverses our technique generalizes in a more fine-grained way, and can be combined with MCMC to provide unbiased samples. Adaptive MCMC techniques such as those presented in Roberts and Rosenthal (2009) and Haario et al. (2006) are used to tune parameters of MCMC algorithms, but do not allow arbitrarily close adaptation of the underlying model to the posterior, whereas our method is designed to allow such close approximation. A number of adaptive importance sampling algorithms have been proposed for Bayesian networks, including Shachter and Peot (1989), Cheng and Druzdzel (2000), Yuan and Druzdzel (2012), Yu and Van Engelen (2012), Hernandez et al. (1998), Salmeron et al. (2000), and Ortiz and Kaelbling (2000). These techniques typically learn Bayes nets which are directed "forward", which means that the conditional distributions must be learned from posterior samples, creating a chicken-and-egg problem. Because our trained model is directed "backwards", we can learn from both prior and posterior samples. Gibbs sampling and single-site Metropolis-Hastings are known to converge slowly in the presence of determinism and long-range dependencies. It is well-known that this can be addressed using block proposals, but such proposals typically need to be built manually for each model. In our framework, block proposals are learned from past samples, with a natural parameter for adjusting the block size.

## 7 Conclusion

We have described a class of algorithms, for the setting of amortized inference, based on the idea of learning local stochastic inverses—the information necessary to "run a model backward". We have given simple methods for estimating and using these inverses as part of an MCMC algorithm. In exploratory experiments, we have shown how learning from past inference tasks can reduce the time required to estimate quantities of interest. Much remains to be done to explore this framework. Based on our results, one particularly promising avenue is to explore estimators that initially generalize quickly (such as regression), but back off to a sound estimator as the training data grows.

**Acknowledgments**

We thank Ramki Gummadi and anonymous reviewers for useful comments. This work was supported by a John S. McDonnell Foundation Scholar Award.

## Footnotes

[1] In a setting where we only ever resample up to $k$ variables, we only need to estimate the relevant inverses, i.e., not all conditionals for the full inverse graph.

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
