[Reviews · NeurIPS 2013]

Submitted by Assigned_Reviewer_6

This paper considers learning to sample from the posterior distribution of a model, by directly predicting latent variables from data. The idea is tested in the block MCMC context, where a small block of latents are predicted from the current state of other latents (and the data). This is shown to perform better than single-site Gibbs when variables are highly correlated and there is sufficient data to train the predictors.

The paper is well written and has a reasonable evaluation. As a concrete inference algorithm, it doesn't convincingly advance the state of the art in Bayesian networks. The comparison between block MCMC and single-site Gibbs is unsurprising. For the networks and block sizes used here, you could compute the exact predictive distribution for each block and sample from it, rather than learning a predictor. An efficient block Gibbs sampling scheme was already proposed by Hamze & de Freitas (2004), using much larger blocks than the ones here.

"From Fields to Trees: On blocked and collapsed MCMC algorithms for undirected probabilistic graphical models"
Firas Hamze, Nando de Freitas, UAI'04
http://www.cs.ubc.ca/~nando/papers/tree2.pdf

Nevertheless, the idea of predicting latent variables from data could be useful for other problems where it is not so easy to sample from blocks. The idea of structuring the prediction function according to the graph structure of the model is a good one, although it's likely that one could improve upon the particular ordering chosen by Algorithm 1.

For networks with high determinism, the block shapes should follow the determinism. But Algorithm 1 determines the shape of the blocks purely via graph structure instead of actual correlations (path length is used as a proxy for correlation).

In Algorithm 4, p_theta is not defined. This should be the unnormalized true density (not involving theta), and it should be in the log domain.

Section 5 should explain why interaction terms didn't help in figure 6, or just exclude this curve.

Does figure 4 hold the total number of training samples fixed? Or does each new task add more training samples?

Section 5 says that "For training and testing, we use the evidence provided along with each network." Does this mean that the network was tested with the same evidence it was trained on?
Summary: An interesting idea for inference but only tested in a limited scenario that existing methods can already handle.

Submitted by Assigned_Reviewer_7

This paper is based on the observation that while doing posterior sampling in
a Bayesian network, information can be collected about conditional relationships
between variables that follow structure that is different from the given
Bayes net. The idea is then to use this insight to learn conditional relationships
for alternative Bayes net factorizations. This is useful either to (a) sample
from these approximate alternative factorizations to get fast approximate samples
by conditioning on the observations then sampling outwards following the directed
structure of the alternative graph, or (b) using the learned conditional relationships to create proposals for Metropolis Hastings MCMC.

The primary benefit of this setup is in the setting of amortized inference, where
it is assumed that we are required to solve many posterior inference tasks over
the same model but where the pattern of observations may change.

There are three main steps:
1. Construct the inverse factorization networks.
2. Learn the parameters for the inverse factorization networks from
posterior sampling runs.
3. Use the learned parameters to improve inference.

The paper gives a heuristic algorithm for (1). Given the justification
provided in the paper, (2) is relatively straightforward, and the paper
suggests either using empirical frequency counts or learning an approximation
via e.g. logistic regression. One inverse network is learned per variable.
For (3), the paper uses the inverse networks
to produce proposals for Metropolis Hastings MCMC. Block proposals
can be generated by proposing jointly the assignments to the "last" k
variables in the inverse network.

Overall, I found the paper to have interesting ideas and to be an enjoyable
read. The exposition is good and the ideas come through clearly.

One work that I'd like to see cited and discussed is data driven MCMC:
Tu, Zhuowen, and Song-Chun Zhu. "Image segmentation by data-driven Markov chain Monte Carlo." Pattern Analysis and Machine Intelligence, IEEE Transactions on 24.5 (2002): 657-673.

Another concern is in how large the parent sets will be in the learned
inverse networks. Specifically, on line 12 of Algorithm 1, we must choose
a parent set that d-separates a variable from the rest of the graph. It
would be nice to have some discussion about how large these sets are in
practice (relatedly, some more motivation about what is going on in Algorithm 1,
what tradeoffs were considered when developing the heuristic, etc. would
be welcome).

A related question is if keeping the parent sets as small as possible
is important. It seems like the extreme opposite choice for the
inverse factorizations would be something like the NADE model [1], which
assumes no conditional independence. Have you experimented with different
inverse factorization algorithms, and can you say what aspects of Algorithm 1
are most important?



[1] The Neural Autoregressive Distribution Estimator
Hugo Larochelle and Iain Murray,
Artificial Intelligence and Statistics, 2011
Summary: Interesting idea with clear exposition. Not 100% confident on the novelty.

Submitted by Assigned_Reviewer_8

This paper presents a method for learning 'recognition networks', i.e., graphical models describing the posterior over hidden nodes given observed ones, whose purpose is to accelerate inference after the model has been learned. The recognition network is a combination of local models, whose parameters are learned using data sampled from the original model, and whose structure is learned by a MCMC scheme. The idea is quite simple and not new, and is therefore mostly evaluated empirically, a job the authors do quite thoroughly (on relatively small networks). I'm curious about the range of applicability of this method and ways to expand this range. For instance, in models where all hidden nodes are correlated by the posterior, no recognition network of the type discussed here may be sufficiently close to the exact posterior; however, if the authors allow networks with additional nodes not in the original model, this problem may be reduced. Also, deep learning uses a recognition network (of a different type, of course) which takes as input all the nodes (hidden and visible), so performance on discriminative tasks is as fast as here, but (I'm guessing) might be better.

Quality: the paper is technically sound.
Clarity: the paper is mostly well written, but I found the section on learning structure harder to follow.
Originality: the idea is not new, the proposal here is richer from previous versions and may be more powerful.
Significance: I imagine the set of people who would use this method is small.
Summary: This is a mostly well written paper which proposes an improved way of learning a separate model of the posterior over hidden units from data sampled from the original model. An empirical test of the method is presented.
Author Feedback

Author rebuttal: We thank the reviewers for their time and effort.

We first address the novelty and significance of our contribution, then respond to individual points.

As the field of machine learning tackles more complex problems, we expect a move away from algorithms for one-off problem-solving and towards systems that interact with and learn from the world over longer time scales. We highlight amortized inference as a critical--but so far neglected--component of such systems, and initiate the development of algorithms explicitly targeted at this setting.

We develop the theory of amortized inference, and describe how it can be used to improve existing sampling algorithms. In contrast to previous work, we show how to learn from both prior and posterior samples. Learning from prior samples allows us to learn in the presence of strong determinism. Learning from posterior samples allows us to cover parts of the state space that have low probability under the prior.

Our goal is *not* to provide a new inference algorithm that can solve problems that existing methods cannot solve. Rather, we exhibit a general-purpose method (applicable to any Bayes net, discrete or continuous) for taking existing samplers and building a fast recognition model. Sharing between different posteriors is key in the setting of amortized inference, but a priori it is unclear whether this is even possible in a compositional fashion. We answer in the affirmative and exhibit an algorithm that transitions from learning a Gibbs sampler to learning a full posterior sampler as training data grows.

We reply to individual points below:

> One work that I'd like to see cited and discussed is data driven MCMC

> An efficient block Gibbs sampling scheme was already proposed by Hamze & de Freitas (2004), using much larger blocks than the ones here.

Existing work on blocked and data-driven MCMC sampling is mostly complementary to our method. Instead of generating training data from the prior or by Gibbs sampling, one could use more advanced algorithms as a data source. Once the trained inverse model performs better than the non-adaptive algorithm (i.e., exceeds some criterion that depends on both sample quality and speed), we switch over to using the trained model. There is much room for future work on the dynamics of training (how long does it take for training to pay off?) and on deeper integration between generating and using training samples (when can we bootstrap by learning from samples generated by a partially trained inverse sampler?).

> The idea of structuring the prediction function according to the graph structure of the model is a good one, although it's likely that one could improve upon the particular ordering chosen by Algorithm 1.

We agree. We view the framework of learning inverses as our main contribution, with the particular instantiations of inverse factorization algorithm, prediction functions, and (MCMC) sampling as illustrative (and useful) examples.

> The idea is quite simple and not new, and is therefore mostly evaluated empirically, a job the authors do quite thoroughly (on relatively small networks).

While the idea does have precursors, we argue that it is a substantial advance, conceptually (setting of amortized inference), theoretically (learning from prior and posterior, sharing across posteriors), and empirically (broad evaluation).

> in models where all hidden nodes are correlated by the posterior, no recognition network of the type discussed here may be sufficiently close to the exact posterior; however, if the authors allow networks with additional nodes not in the original model, this problem may be reduced.

Introducing additional nodes in the inverse network is an interesting idea that we hope to see explored in future work.

However, note that our method can always exactly represent the posterior (using the empirical frequency estimator). We prove that, as training data set size goes to infinity, we converge to a perfect posterior sampler. The purpose of additional hidden variables would be to speed up learning.

> The recognition network is a combination of local models, whose parameters are learned using data sampled from the original model, and whose structure is learned by a MCMC scheme.

There seems to be a misunderstanding: The inverse model itself is not learned using MCMC. We deterministically compute the structure of the inverse model based on the structure of the forward model, and the parameterization using fast estimators.

> Another concern is in how large the parent sets will be in the learned inverse networks.

We have experimented with approximate inverse factorizations. For a bipartite disease-symptom Bayes nets, restricting the size of the parent sets resulted in effects similar to replacing the frequency estimator with logistic regression: i.e., faster learning at the cost of converging to a less accurate model.

> Section 5 should explain why interaction terms didn't help in figure 6, or just exclude this curve.

Thanks for pointing out this mislabeling! L2 with and without interactions are switched. As described in the main text, interactions did help.

> Section 5 says that "For training and testing, we use the evidence provided along with each network." Does this mean that the network was tested with the same evidence it was trained on?

For the in-depth evaluation, we used different evidence for training and testing. For the broad evaluation, we used the same evidence in order to remain close to the original challenge, which provides only one setting for the evidence variables.

> Does figure 4 hold the total number of training samples fixed?

Each task adds more training samples. We will clarify.